# How the insect central complex could coordinate multimodal navigation

**Xuelong Sun[1,2]\*, Shigang Yue[1,2]\*[†], Michael Mangan[3]\*[†]**

[1]Machine Life and Intelligence Research Centre, School of Mathematics and Information Science, Guangzhou University, Guangzhou, China; [2]Computational Intelligence Lab and L-CAS, School of Computer Science, University of Lincoln, Lincoln, United Kingdom; [3]Sheffield Robotics, Department of Computer Science, University of Sheffield, Sheffield, United Kingdom

**Abstract** The central complex of the insect midbrain is thought to coordinate insect guidance strategies. Computational models can account for specific behaviours, but their applicability across sensory and task domains remains untested. Here, we assess the capacity of our previous model (Sun et al. 2020) of visual navigation to generalise to olfactory navigation and its coordination with other guidance in flies and ants. We show that fundamental to this capacity is the use of a biologically plausible neural copy-and-shift mechanism that ensures sensory information is presented in a format compatible with the insect steering circuit regardless of its source. Moreover, the same mechanism is shown to allow the transfer cues from unstable/egocentric to stable/geocentric frames of reference, providing a first account of the mechanism by which foraging insects robustly recover from environmental disturbances. We propose that these circuits can be flexibly repurposed by different insect navigators to address their unique ecological needs.

**\*For correspondence:**
xsun@lincoln.ac.uk (XS);
syue@lincoln.ac.uk (SY);
m.mangan@sheffield.ac.uk (MM)

[†]These authors contributed equally to this work

**Competing interest:** The authors declare that no competing interests exist.

## Editor's evaluation

This *eLife* Advance by Sun et al. expands on a previous publication in 2020 which developed and outlined a model for biologically inspired visual navigation circuit. Here they include odour and wind input and successfully recreate complex multimodal behavioural observations in ants, illustrating that the model is suited to generate viable hypotheses about circuit-level implementation of navigational control networks in insects in response to varied sensory inputs.

## Introduction

Recently, it has been proposed that the repertoire of robust navigation behaviours displayed by insects (*Webb and Wystrach, 2016*; *Wehner, 2019*) can be traced to the well-conserved brain region known as the central complex (CX) (*Honkanen et al., 2019*; *Hulse et al., 2021*). The evidence to support this hypothesis includes the discovery of the insect head-direction system in the CX that tracks the animal's current heading relative to external (*Heinze, 2014*; *Seelig and Jayaraman, 2015*; *Kim et al., 2019*; *Hardcastle et al., 2021*) or self-motion (*Green et al., 2017*; *Turner-Evans et al., 2017*) cues; the innervation of the fan-shaped body (FB) region of the CX with sensory information relevant to different orientation strategies (*Hu et al., 2018*; *Franconville et al., 2018*; *Hulse et al., 2021*; *Shiozaki et al., 2020*); the well-preserved columnar structure that is well suited to computing desired headings for vector navigation tasks (*Stone et al., 2017*; *Honkanen et al., 2019*; *Le Moël et al., 2019*; *Lyu et al., 2020*); and the identification of a neural steering circuit in the FB capable of computing motor commands that reduce the offset between the current heading and a desired heading (*Stone et al., 2017*; *Honkanen et al., 2019*;

*Rayshubskiy, 2020*). Computational models of this architecture have produced realistic path integration (PI) (*Stone et al., 2017*; *Gkanias et al., 2019*) and trap-lining behaviours (*Le Moël et al., 2019*), and simple conceptual extensions have been outlined that could account for long-distance migratory behaviour (*Honkanen et al., 2019*). Yet, for the CX to be considered a general navigation centre, it must additionally be capable of (i) generating gradient ascent/descent behaviours that rely on spatially varying but rotationally invariant sensory cues (e.g. odour gradients), (ii) coordinating competing guidance systems into a single meaningful motor command, and (iii) generalise across sensory modalities and task spaces.

We recently demonstrated how the steering circuit could be adapted to ascent gradients of visual familiarity when augmented by a neural '*copy-and-shift*' mechanism that converts temporal changes in spatially sampled sensory information into an orientation signal (*Sun et al., 2020*). Specifically, the mechanism firstly *copies* the animal's current heading from the head-direction cells in the protocerebral bridge (PB) to desired heading networks in the FB. At the same time, the signal undergoes a lateral *shift* in proportion to any undesired change in sensory valence as measured by the mushroom body (MB) output neurons (*Aso et al., 2014*; *Li et al., 2020*; *Hulse et al., 2021*). Thus, the animal will continue on its current heading until an undesirable change in sensory valence is experienced at which point the shift mechanism will create an offset between the current and desired headings, causing the steering circuit to initiate a change of direction. The architecture of the CX already possesses neural substrates ideally suited for both the '*copy*' and '*shift*' functions: head-direction cells are known to transmit their output into the ring structures of the central body (*Stone et al., 2017*; *Honkanen et al., 2019*) as needed for *copy* stage; and neural mechanisms that laterally shift the head-direction cells in response to sensory feedback (e.g. the self-motion cues [*Turner-Evans et al., 2017*; *Green et al., 2017*], the visual cues [*Kim et al., 2019*; *Fisher et al., 2019*]) are well established as required for the *shift* stage. Crucially, the complete '*copy-and-shift*' mechanism explains how the CX steering circuit (see *Figure 1*) could exploit sensory gradients that provide no instantaneous orientation information for navigation.

We also demonstrated neural mechanisms that coordinate between different guidance strategies (*Sun et al., 2020*). Specifically we added a contextual-switching mechanism (see *Figure 1*) that triggers specific guidance strategies depending on the context, for example, switching from PI unfamiliar surroundings to visual route following in familiar terrain. As a final stage, we revealed how ring attractor (RA) circuits (*Touretzky, 2005*; *Sun et al., 2018*; see *Figure 1*) that we hypothesise exist in the FB provide an ideal substrate for optimally integrating cues that exist within a shared context (e.g. PI and visual homing [VH] in unfamiliar contexts). The '*copy-and-shift*' mechanism again plays a crucial role in this capacity as it 'transfers' orientation outputs into a shared frame of reference. For example, when ascending gradients, temporal changes in visual familiarity are translated into heading commands relative to the head-direction system which then share a frame of reference with the PI system.

This biologically constrained model of the insect midbrain was shown to be capable of generating realistic visual navigation behaviours of desert ants through the coordinated action of visual route following (RF), VH, and PI modules partially addressing two of the requirements listed above (*Sun et al., 2020*). In this study, we extend our analysis of the model, and in particular the '*copy-and-shift*' mechanism, to assess if it can address the latter issue of generalisation across and between sensory and task domains. The following sections first assess whether the model can be easily reapplied to the olfactory tasks of chemotaxis and odour-gated anemotaxis (plume following) in laboratory-like settings. We then probe whether the same integration mechanisms can generalise to odour-gated switching in both flies and desert ants. Finally, we provide the first account of how the CX could transfer orientation cues from an egocentric to a geocentric frame of reference which we propose can enhance the robustness of navigation.

## Results

### Core odour navigation behaviours using *copy-and-shift*

Here we assess the ease with which our visual navigation model generalises olfactory navigation tasks.

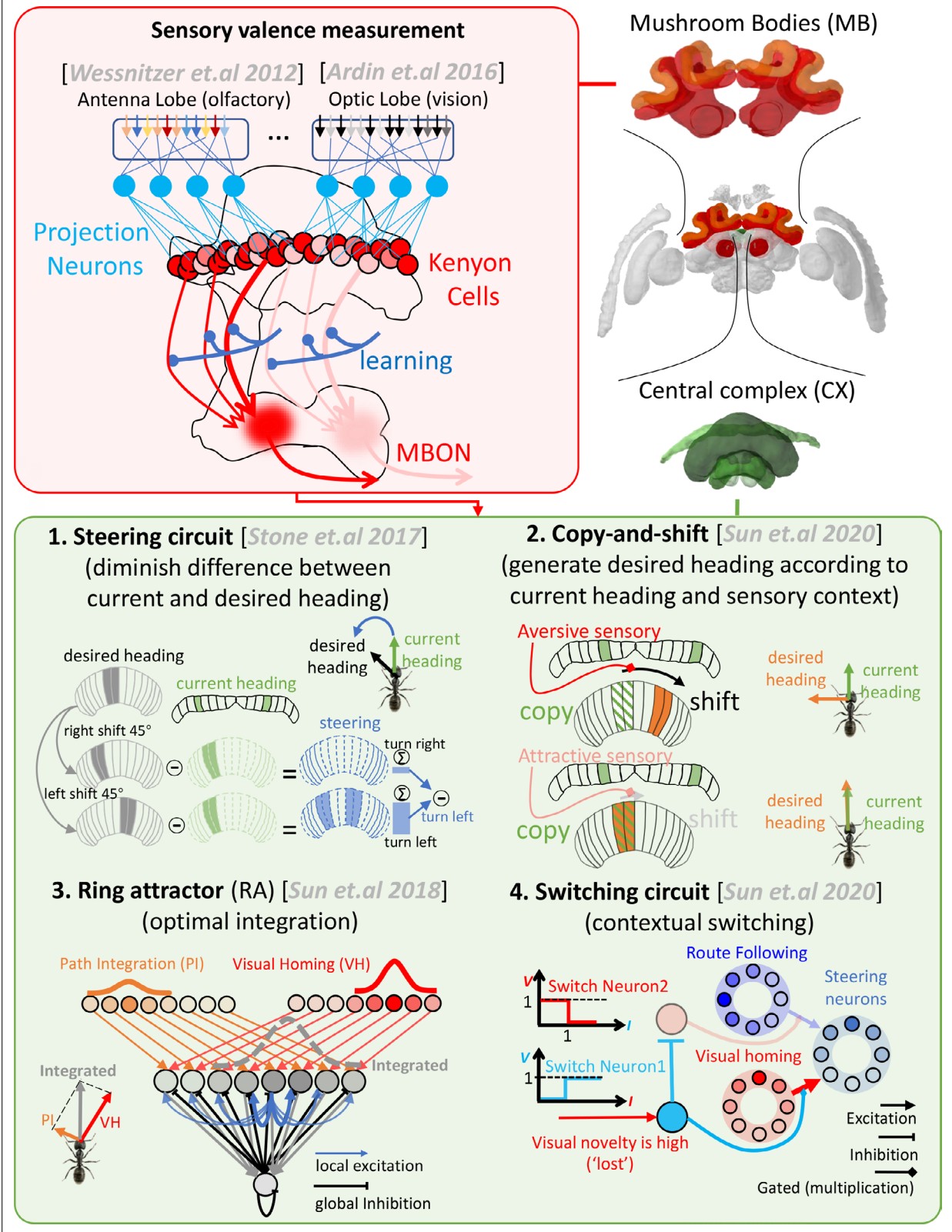

**Figure 1.** Schematic overview of the mushroom body-central complex (MB-CX) model first presented in **Sun et al., 2020** and reapplied here to multimodal guidance. The upper-right panel depicts the two key brain areas modelled (MBs in red, central in green). The upper-left panel (red background) outlines the role of the MBs in measuring valence of odour (**Wessnitzer et al., 2012**) and visual (**Ardin et al., 2016**) cues. The lower panel (green background) introduces the four CX subcircuits integrated in the previous model. (1) The steering circuit proposed to exist in the CPU1 neurons

*Figure 1 continued on next page*

*Figure 1 continued*

of the CX that computes the turning angle that minimises the difference between the current heading (from the protocerebral bridge [PB]) and desired heading (from the CPU4 cells) (*Stone et al., 2017*). (2) The copy-and-shift mechanism creates a desired heading from non-directional cues by simply copying the current heading and shifting it in proportion to the change in sensory valance. (3) Ring attractor networks can automatically and optimally integrate orientation cues from disparate sources into a single readout. Our model uses RAs to integrate both compass and desired heading signals. (4) Context-dependent switches multiplex systems at a high level (e.g. when 'lost' engages visual homing [VH] but not route following [RF]). Images of the brain regions are adapted from the Insect Brain Database (*Heinze et al., 2021*; https://www.insectbraindb.org).

## Chemotaxis of odour gradients

Adult and larvae fruit flies readily climb rewarding odour gradients by modulating their heading direction in direct response to the temporal change in odour concentration (*Gomez-Marin et al., 2010*; *Nagel and Wilson, 2011*; *Kim et al., 2011*; *Schulze et al., 2015*; *Jung et al., 2015*) mirroring our model's approach to VH. Moreover, the neural pathways of olfactory processing are well established and only differ from our model in their sensory origins (antennal lobe [AL] to the lateral horn [LH] [*Gupta and Stopfer, 2012*; *Roussel et al., 2014*] and MBs [*Aso et al., 2014*; *Hulse et al., 2021*]) before connecting to the CX through direct or indirect (hypothetically via superior medial protocerebrum [SMP]; *Plath et al., 2017*; *Hulse et al., 2021*; *Li et al., 2020*) neural pathways. Thus by simply changing the input from optic to ALs and the processing region from the MB to the LH and MB (see *Figure 2A*, left panel) our model is able to adapt its heading to align with the positive odour gradient over successive steps (see *Figure 2B*, left panel). Note that here we simply take the valence output of the MB as the odour concentration, buy any other equivalent measurement (such as the degree of attraction) could work along with the 'copy-and-shift' mechanism. *Figure 2C* (left panel) demonstrates the realistic chemotaxis behaviour generated by the model in a classic 'volcano' environment (*Jung et al., 2015*; *Schulze et al., 2015*). *Figure 2—figure supplement 1* provides similarly realistic paths in other odour landscapes. It should be noted that there are neural pathways not included in the model that directly link odour input to motor outputs that may play a role in chemotactic guidance (*Green et al., 2019*; *Rayshubskiy, 2020*; *Scaplen et al., 2021*). Indeed while larvae possess MB and LH assemblies, they do not have a fully developed CX as modelled here (*Gowda et al., 2021*). Analysis of behavioural deficiencies in animals with CX knockouts would offer crucial insights into the role of the CX for chemotactic behaviours.

## Anemotaxis in odour plumes

In moving airflows, adult fruit flies pinpoint olfactory sources by anemotaxis whereby individuals align with the upwind direction, allowing them to approach the hidden odour source (*Kennedy and Marsh, 1974*; *Rutkowski et al., 2009*; *van Breugel and Dickinson, 2014*). Insects sense wind direction through deflections of their antennae (*Yorozu et al., 2009*; *Patella and Wilson, 2018*; *Okubo et al., 2020*) with the wedge projection neurons (WPNs) converting their inputs (via antennal mechanosensory and motor centre [AMMC] pathway in *Figure 2B*, right panel) into a direction relative to the animal's current heading (*Suver et al., 2019*; see *Figure 2—figure supplement 2*). The WPN output is then transmitted to the FB of the CX via the lateral accessory lobe (LAL) -> noduli (NO) pathway (*Hulse et al., 2021*; *Matheson et al., 2021*; *Figure 2B*, right panel). The '*copy-and-shift*' mechanism again provides the ideal bridge between input signal and steering circuit. By simply driving the direction and magnitude of the '*shift*' by the WPN response when a rewarding odour is detected (*Figure 2A*, right panel), the model turns the agent upwind (see *Figure 2B*, right panel). *Figure 2C* (right panel) shows an example path of a simulated fly navigating a classic laboratory environment with an odour plume into which rewarding odour is toggled ON and OFF (for a simulation of a group agents, see *Figure 2—figure supplement 3*), which demonstrates realistic odour-driven anemotaxis behaviour.

Taken together the above data demonstrate the capacity of the model to generalise from visual to olfactory navigation without significant alteration.

## Coordination of guidance behaviours by linking frames of reference

With the model shown to generalise from visual to olfactory navigation tasks, we now assess its ability to coordinate guidance strategies across sensory domains.

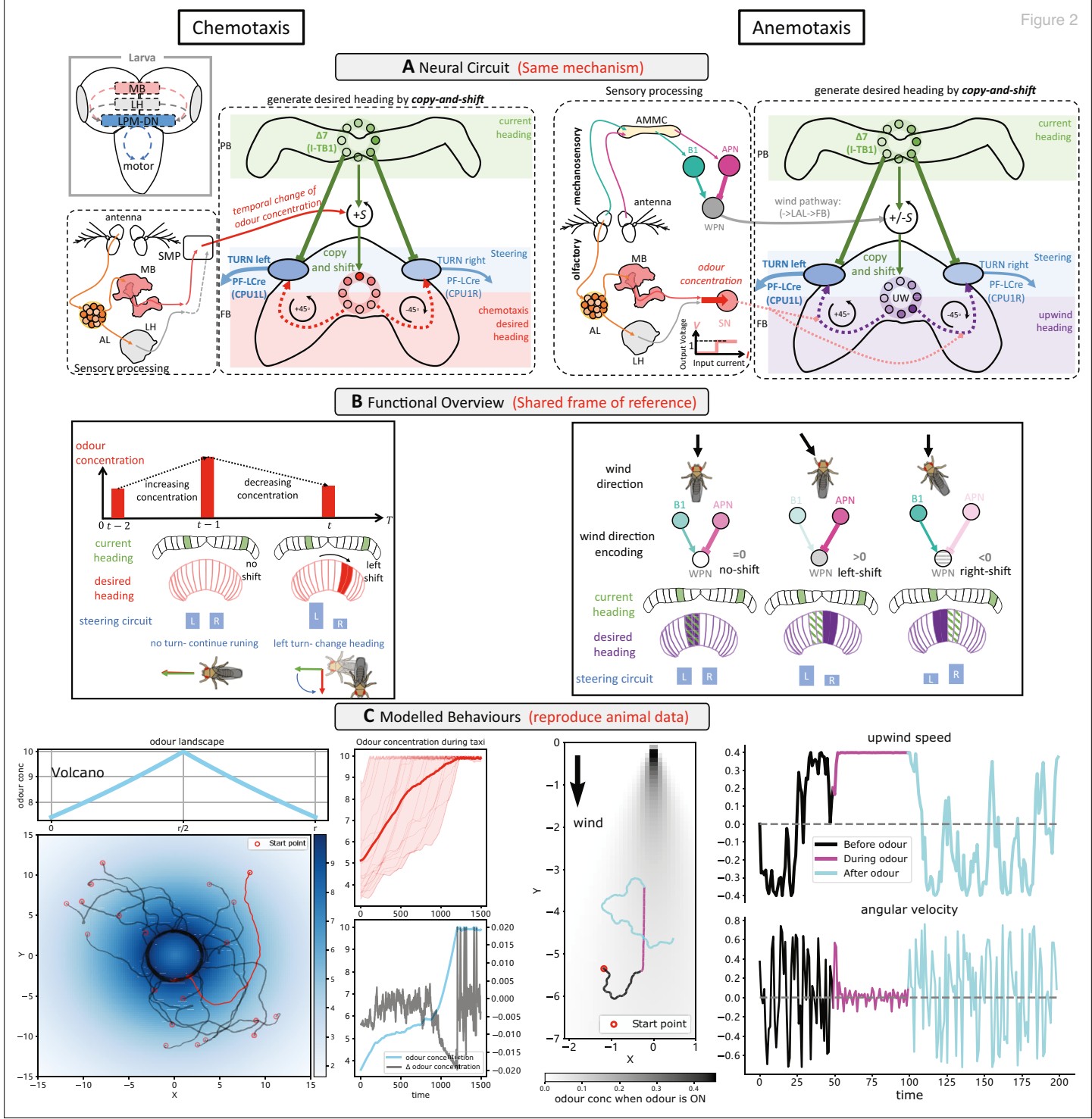

**Figure 2.** Modelling olfactory navigation in flies using a 'copy-and-shift' mechanism: chemotaxis (left side) and anemotaxis (right side). (**A**) Schematic diagrams of the neural circuits generating current-desired heading pairings for chemotaxis and anemotaxis. The copy-and-shift mechanism is only different in how the shift is realised: for chemotaxis, the temporal change of the odour concentration produces turns of different magnitude in a predefined direction, which for anemotaxis the wedge projection neuron (WPN) provide both turning magnitude and direction to steer the animal upwind. The corresponding hypothesised functional map of larvae brain is inserted in the left panel, showing that the olfactory descending neurons LPM-DN may play similar role as the central complex (CX). (**B**) Schematic diagram explaining the model functions. For chemotaxis, decreasing odour concentration will shift the desired heading from current heading causing the steering circuit to initiate a turn. For anemotaxis, the WPNs subtract the activation of the antennal mechanosensory and motor centre (AMMC) projection neuron (APN) from that of B1 that directly shifts the desired heading

*Figure 2 continued on next page*

*Figure 2 continued*

to align with the upwind direction. Note that the two mechanisms share a frame of reference. (**C**) Example behaviours generated by the model. Realistic chemotaxis behaviour is shown (left) in a 'volcano' odour landscape. On the right, realistic anemotaxis (magenta path segment) are shown when odour is 'ON' vs. undirected motion (black and cyan path segments) when odour is 'OFF'. Upwind speed and angular velocity of the example agent are shown on the right panel. Note the obvious higher upwind translational velocity and low angular velocity during the presence of the odour indicate surges upwind.

The online version of this article includes the following figure supplement(s) for figure 2:

**Figure supplement 1.** The simulation results of chemotaxis model with odour landscape of 'linear'.

**Figure supplement 2.** Simulation of wind direction encoding.

**Figure supplement 3.** Simulation results of a group of agents (N = 20) driven by the odour-driven anemotaxis model.

## Contextual switching between olfactory guidance behaviours

In reality insects utilise both the chemotaxis and anemotaxis strategies outlined above. Across species and environments (laminar odour gradient or turbulent odour plume), a distinct behavioural trigger is reported at the onset (ON response) or loss (OFF response) of sensory valence (moths [***Kennedy and Marsh, 1974***; ***Rutkowski et al., 2009***], flying fruit flies [***van Breugel and Dickinson, 2014***], walking flies [***Steck et al., 2012***; ***Bell and Wilson, 2016***; ***Álvarez-Salvado et al., 2018***]). Specifically, in the presence of the attractive odour animals apply anemotaxis and surge upwind, but when the attractive odour is lost they engage in a chemotactic-like search to recover the plume. This problem is analogous with the contextual switching used in our previous model to select between ON- and OFF route navigation strategies (***Wystrach et al., 2012***). ***Figure 3A*** (left panel) depicts how the CX switching circuit can be easily reconfigured to be triggered by the instantaneous change of odour concentration fitting with the reported ON and OFF responses (***Álvarez-Salvado et al., 2018***). Note that we here assume that the ON and OFF responses are driven by the output neurons of the odour processing brain regions (i.e. MBON or LHON) that could compute the temporal changes of odour concentration (***Dolan et al., 2018***; ***Hulse et al., 2021***; ***Matheson et al., 2021***). ***Figure 3B*** (left panel) illustrates simulated ON and OFF responses that are supplied to the model and their behavioural consequence. ***Figure 3*** (left panel) demonstrates realistic olfactory navigation behaviour similar to the behavioural data in ***Álvarez-Salvado et al., 2018***. See also the simulation results of a 20-agent group demonstrating similar performance in ***Figure 3—figure supplement 1***.

## Optimally integrating navigation behaviours across sensory domains

In barren salt pans, homing desert ants follow their path integrator to their nest area before relying on nest odour plumes for their final approach (***Buehlmann et al., 2012***). Ants bypass the nests of conspecifics that diffuse similar odours ($CO_2$) until reaching the nest locale (***Buehlmann et al., 2012***), indicating use of a sophisticated integration strategy beyond simple switching outlined above. Rather, ants instead appear to weigh their PI output relative to the home vector length in a similar fashion to their integration of PI and visual cues (***Wystrach et al., 2015***; ***Legge et al., 2014***) as was realised in our previous model using ring attractor networks (***Touretzky, 2005***; ***Sun et al., 2018***; ***Sun et al., 2020***). ***Figure 3A*** (right panel) depicts the augmentation of our odour-gated anemotaxis model with a ring attractor circuit to optimally integrate PI and olfactory navigation outputs. These adaptations are in accordance with the olfactory navigation mechanisms (chemotaxis and anemotaxis) proposed to be used by ants by ***Wolf and Wehner, 2000***; ***Wolf and Wehner, 2005***. Note that the desired headings recommended by odour homing (OH, or chemotaxis) and upwind direction (UW, or odour-gated anemotaxis) are gated by the OFF and ON responses and weighted by the odour concentration signal prior to being injected into the ring attractor to be combined with PI. ***Figure 3B*** (right panel) illustrates how the various desired heading signals are optimally integrated by the ring attractor network before being sent as input to the steering circuit. ***Figure 3C*** shows homing paths generated by the model following simulated displacements left or right of the regular feeder which closely match those of real ants (***Buehlmann et al., 2012***). Note that there is an additional odour plume diffused by a simulated conspecific nest positioned near the release points which causes some distraction before the simulated ants continue to the real nest site. In the absence of the distractor, nest paths are much more direct (see ***Figure 3—figure supplement 1***).

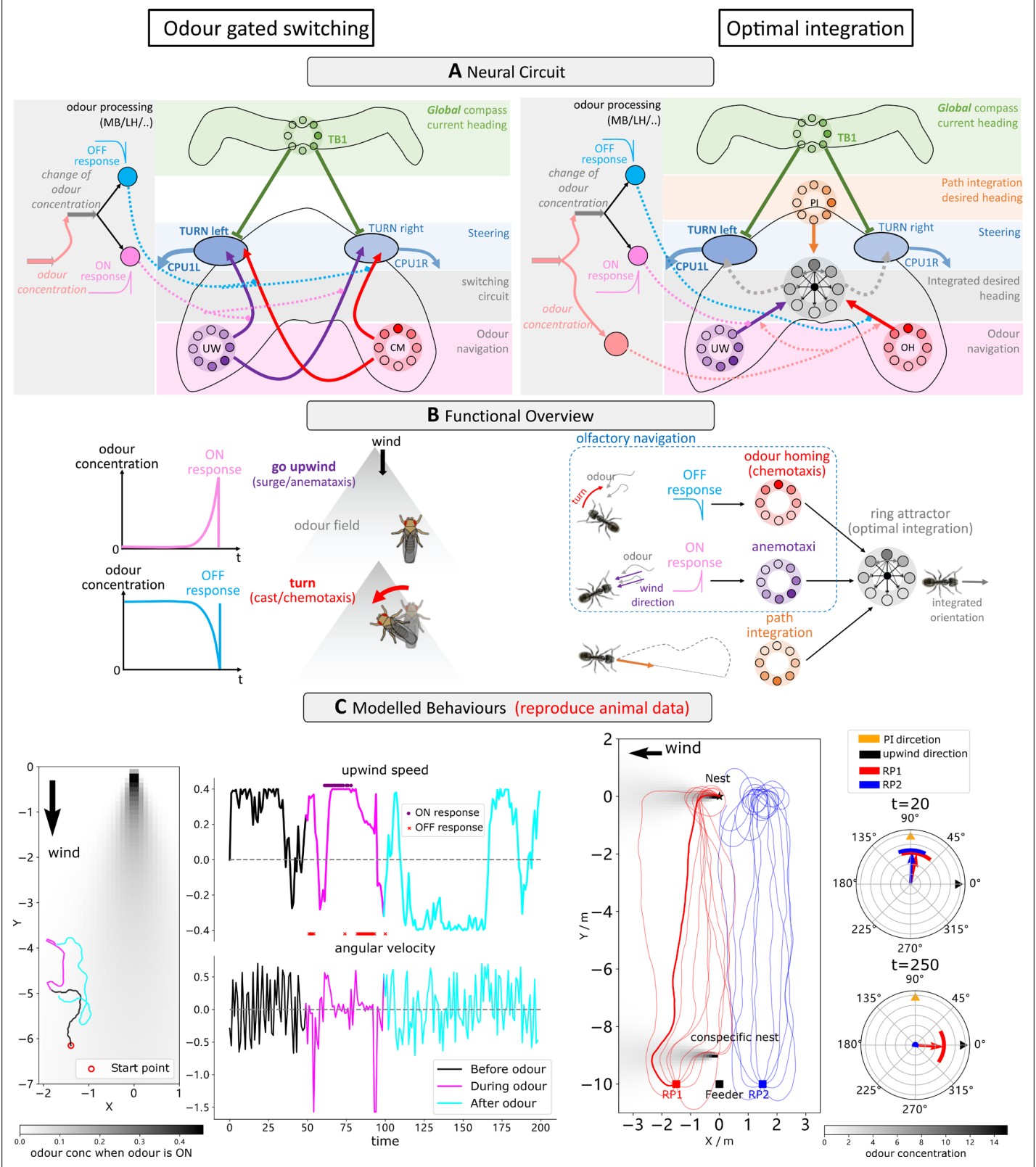

**Figure 3.** Optimal coordination of guidance behaviours that share a frame of reference. (**A**) Schematic diagrams of the integration circuits. Left: temporal change in odour concentration based ON and OFF responses drives the switching circuit to select between chemotaxis or anemotaxis strategies. Right: ring attractor network integrates multiple cues weighted by sensory valence. (**B**) Functional explanations of the model. Left: ON responses trigger upwind turns while OFF responses trigger chemotaxis, leading the animal back into the odour plume. Right: ring attractors serve as

*Figure 3 continued on next page*

*Figure 3 continued*

the optimal integration circuit to mediate between anemotaxis, chemotaxis, and path integration systems. (**C**) Example behaviours generated by the model in an anemotaxis and ant homing task. Left part of the left panel shows the trajectory of the one simulated fly, and the upwind speed and angular velocity of the agent are shown in the right part. The time at which ON and OFF responses are triggered are shown by purple dots and red stars, respectively. The left panel of the right-side data shows paths of simulated ants when guided by path integration (PI) and odour cues. Groups headings are also shown at t = 20 (early in the route when PI dominates) and t = 250 (later in the route when olfactory navigation begins to dominate as PI vector length is low).

The online version of this article includes the following video and figure supplement(s) for figure 3:

**Figure supplement 1.** The simulation results of a 20-agent group driven by the ON and OFF responses-based switching model.

**Figure supplement 2.** Sensory perception and neural activities of the highlighted ant driven by the proposed model.

**Figure supplement 3.** Simulation results where there is no conspecific nest near the releasing points with comparison to *Figure 3C*, right panel.

**Figure 3—video 1.** The animation showing the simulation process including homing trajectories, dynamic neural activation, odour measurement, etc.
https://elifesciences.org/articles/73077/figures#fig3video1

Taken together these data demonstrate that the CX possess the neural mechanisms to flexibly coordinate the various guidance behaviours observed in insects across sensory domains, supporting its role as the navigation centre (*Honkanen et al., 2019*; *Hulse et al., 2021*).

## A mechanism for transferring between orientation frames of reference

The optimal integration model detailed above is reliant on the *copy-and-shift* mechanism firstly ensuring that all orientation cues are presented in a shared frame of reference. Recall that the desired headings for PI, chemotaxis and anemotaxis are all defined in relation to the animal's global head direction. In the following analysis, we assess whether this frame-changing capacity can also provide benefits for navigational robustness.

### From egocentric wind direction to geocentric celestial compass

Desert ants travel to and from familiar feeder locations via visually guided routes (*Kohler and Wehner, 2005*; *Mangan and Webb, 2012*), but wind gusts can blow them off course. *Wystrach and Schwarz, 2013* reported that in the instant prior to displacement ants assume a stereotypical 'clutching' pose during which they transfer their egocentric measure of wind direction (indicating the direction in which they are about to be blown) into a geocentric frame of reference given by their celestial compass. Displaced ants then utilise this celestial compass memory to guide their path directly towards their familiar route (*Figure 4A*, left panel). Such a strategy is easily accounted for by the '*copy-and-shift*' mechanism as seen in *Figure 4B* (left panel). That is, during the clutch pose the celestial compass heading is *copied* and *shifted* by the activation of the WPN encoding the upwind direction relative to the animal's heading to create a desired heading that points back along the direction of travel. This desired heading is maintained in a working memory during displacement before activation to guide the agent back to the familiar route region (see simulated navigating paths in *Figure 4C*, left panel).

### From visual context to geocentric celestial compass

Similarly, homing desert ants captured just before entering their nest and released in unfamiliar visual surroundings initially dash back along the celestial compass heading in which they were travelling (*Wystrach et al., 2013*; *Figure 4A*, right panel). Note that this differs from the behaviour of ants lacking PI cues and displaced from other locations along the route. Those ants have no preferred direction of travel following displacement according to the observation (*Wystrach et al., 2013*). This indicates that the sight of the nest surroundings could be considered a 'special circumstance' in a similar way to the 'clutching' pose mentioned above. *Figure 4B* (right panel) depicts how this behaviour could also arise from the '*copy-and-shift*' mechanism. That is, when there is a significant drop of visual novelty (as might only be experienced after a displacement from the nest), the compass direction is again *copied* and *shifted* by a predetermined amount, this case 180°. This creates a new desired heading that can be stored in working memory that will cause the initial search to be focused in the direction from which the animals just travelled (*Figure 4C* , right panel).

In summary, the data above demonstrate the flexibility of the '*copy-and-shift*' mechanism to transfer directional cues from an unstable frame of reference such as the wind direction to a stable frame of

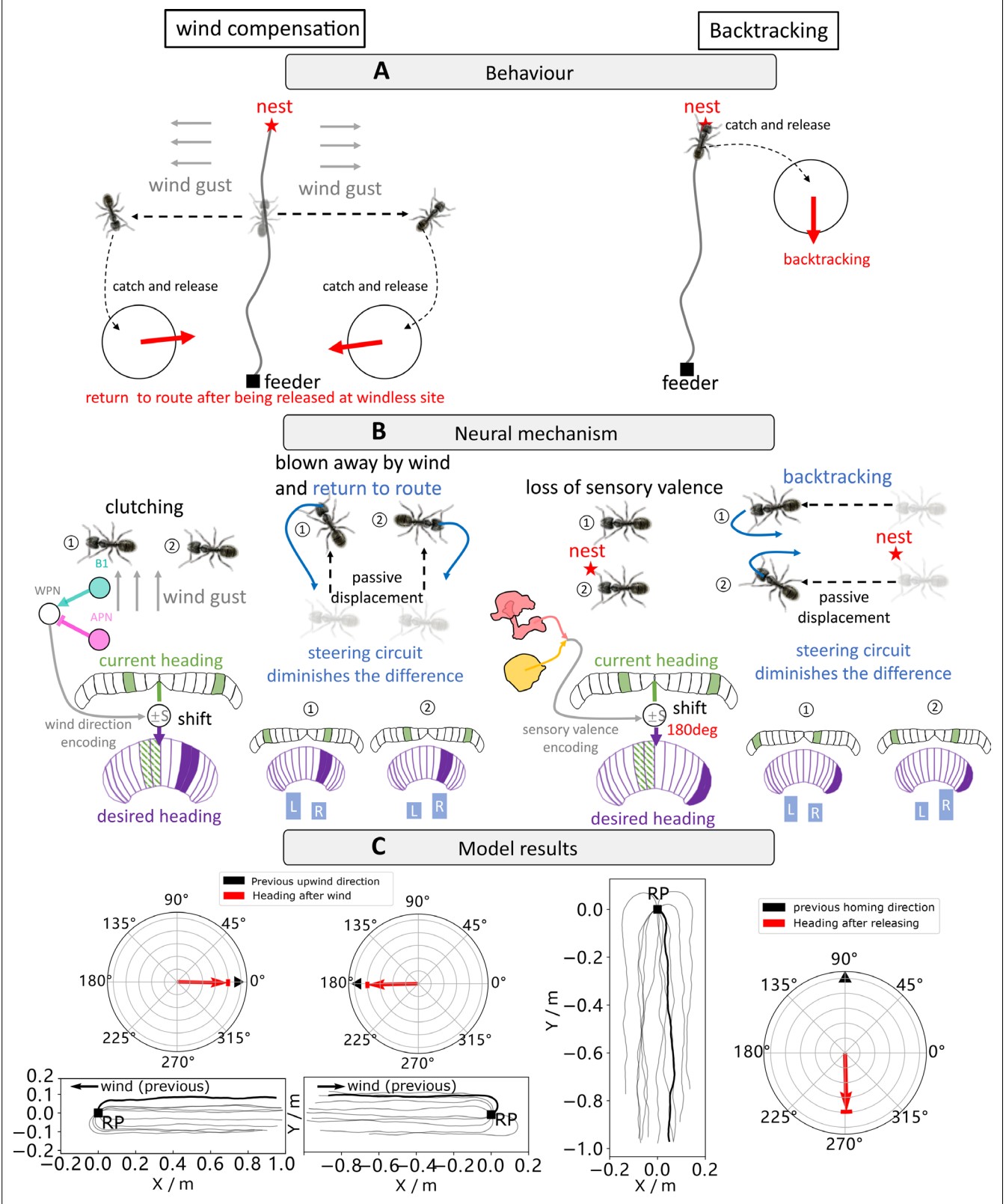

**Figure 4.** Navigating using egocentric and geocentric frames of reference. (**A**) Wind compensation and backtracking behaviour of navigating ants. Left panel illustrates the wind compensation behaviour where ants reorientate to the direction from which they were blown off course but with respect to their celestial compass (**Wystrach and Schwarz, 2013**). Right subfigure shows backtracking behaviours whereby homing desert ants captured just before entering their nest and released in unfamiliar visual surroundings initially dash back along the celestial compass heading in which they were

*Figure 4 continued on next page*

*Figure 4 continued*

travelling (*Wystrach et al., 2013*). (**B**) The proposed neural mechanism showing how the behaviours in (**A**) could be recreated. Wind compensation is implemented by using copy-and-shift to copy their heading compass stored in the central complex (CX) when clutching and shift by an amount of degree determined by the activation of wedge projection neurons (WPNs) to form the working memory (desired heading) for later navigation. Backtracking is modelled in an identical way except that the shift is constant 180°. (**C**) The simulation results of our model. In each panel, the navigating trajectories and initial headings of the simulated ants are shown. Simulated ants guided by the model are all heading to the expected orientation as observed in real behavioural experiments (*Wystrach and Schwarz, 2013*; *Wystrach and Schwarz, 2013*).

reference such as the global celestial compass which can be used at a later time. We proposed that this transfer is triggered by special sensory experience and motivational state of the animal that could be driven by some of the numerous tangential inputs from multiple upstream brain regions to the FB (*Franconville et al., 2018*; *Hulse et al., 2021*) forming a contextually dependent guidance network. This again extends the repertoire of guidance behaviour that the mechanism can account for and further supports the role of the CX as a navigation centre.

## Discussion

To summarise, we have shown how the CX-based steering circuit augmented with a *copy-and-shift* functionality can generate realistic odour-based chemotaxis and anemotaxis behaviours adding to the PI, VH, visual RF, and long-range migrations explained previously (*Stone et al., 2017*; *Honkanen et al., 2019*; *Sun et al., 2020*). We have also outlined CX-based mechanisms that can coordinate guidance cues across sensory domains using biologically realistic context-dependent switches and ring attractor networks. Finally, we demonstrated how the *copy-and-shift* mechanism can facilitate the transfer of orientation cues between unstable to stable frames of references. By triggering such a transfer under specific environmental conditions, insects can increase the robustness of their guidance repertoire. The model presented can thus be considered as a general navigation model extending across multiple behavioural tasks (alignment with rotationally varying compass, visual route, or wind cues; and gradient ascent of spatially varying but rotationally invariant cues such as odour and visual memories) experienced in multiple contexts. Taken together the results add further validation to the claim that the CX acts as the seat of navigation coordination in insects.

The CX is as ancient as insects themselves (*Homberg, 2008*; *Strausfeld, 2009*) and is highly conserved across different species solving different navigational tasks (*Honkanen et al., 2019*; *Hulse et al., 2021*). This fixed circuitry thus appears optimised to receive input from a variety of sensory sources and return a similar variety of navigational behaviours applicable across contexts. Indeed, *Doyle and Csete, 2011* posits that such 'bowtie' (or hourglass) architecture is also observed in the decision-making circuits of the mammalian brain (*Redgrave et al., 1999*; *Humphries and Prescott, 2010*) and function by providing '*constraints that deconstrain*' (see *Figure 5A*). That is, the fixed circuitry of the CX constrains the format of the sensory input but decontrains the application domains of the output behaviours. Through interpreting various navigation behaviours through the lens of the '*copy-and-shift*' mechanism, our model can be considered an example of such bowtie structure within the CX (*Figure 5B*).

This study has explored the behavioural consequences of the mechanisms using abstracted neural implementations, raising the question as to whether they can be realised in insect brains. Regarding the *copy-and-shift* mechanism, lateralised neural connections and synapse plasticity that shift the head-direction output relative to sensory input (i.e. nudge the activation 'bump' within a population of neurons) have already been mapped (*Seelig and Jayaraman, 2015*; *Green et al., 2017*; *Kim et al., 2019*; *Fisher et al., 2019*) and modelled (*Cope et al., 2017*), demonstrating the feasibility of such computation. More recently, *Goulard et al., 2021* presented a CX-based navigation model that includes a biologically realistic neural pathway that is functionally similar to the *copy-and-shift* mechanism proposed here. The same study also outlined how a short-term memory of a desired heading could be maintained in the FB of the CX via synapse-weight modulation after the original guidance cue is removed that could support the wind compensation and backtracking behaviours described above. Our model hypothesises the existence of a ring attractor network to optimally integrate the desired heading cues which we suggest could be realised in the complex intra-connections within the FB and the NO (*Hulse et al., 2021*; *Sayre et al., 2021*). We also hypothesise that different populations

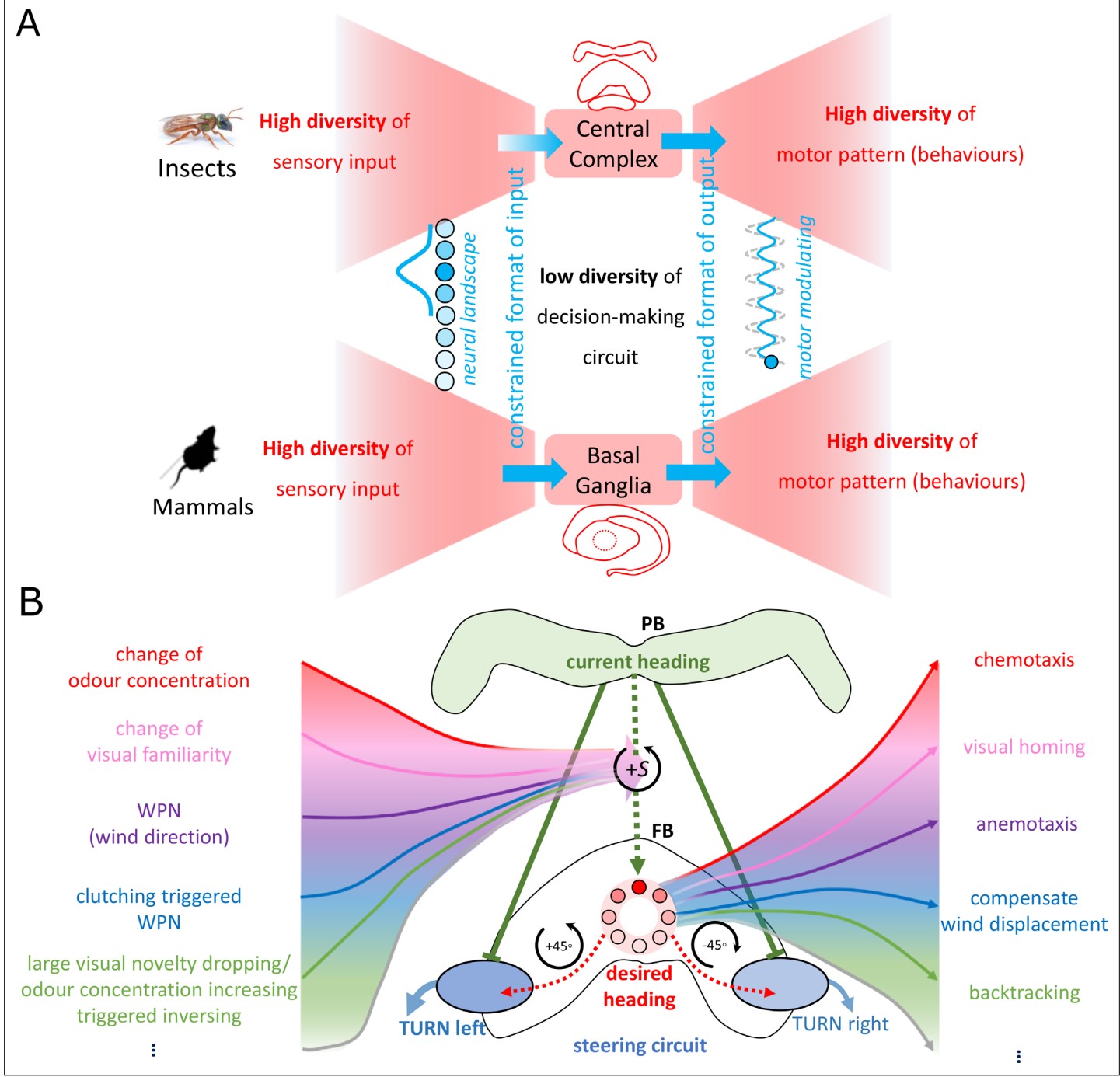

**Figure 5.** The 'bowtie/hourglass' architecture (*Doyle and Csete, 2011*) of biological control system. (**A**) The control systems of insect navigation (top) and mammalian decision-making (bottom) are epitomised by the 'bowtie' architecture, proposing that fixed brain circuitry constrains the format of the sensory input (fanning in to the knot) but decontrains the application domains of the output behaviours (fanning out of the bowtie). Photo of sweet bee *Megalopta genalis* is from Ajay Narendra. (**B**) The proposed mapping of the bowtie architecture to the central complex (CX) for insect navigation. Specially, the copy-and-shift mechanism (regarded as the knot of the bowtie, thus constrains the representation) is reused to generate different desired headings across sensory and task domains (deconstrains the motor pattern, thus allows for high diversity of behaviours).

of Protocerebral bridge - Fan-shaped body - Nodulus (PFN) neurons in the CX simultaneously store the distinct desired headings computed by the independent navigation systems (e.g. PI-based home vector is stored in CPU4 neurons [a subset of PFNs]; *Stone et al., 2017*; *Hulse et al., 2021*; *Sayre et al., 2021*). Further, the hypothetical context switching introduced could be achieved by the recently mapped FB-NOc neurons found in the bees (*Sayre et al., 2021*).

It is also worth noting that the simulated odour perception utilised here is very simplistic. For example, we assume that the odour stimulus (with or without a laminar airflow) forms a stable gradient, which while reflecting the laboratory settings in behavioural studies (*Gomez-Marin et al., 2010*; *Gomez-Marin and Louis, 2012*; *Álvarez-Salvado et al., 2018*) simplifies the spatiotemporally complex plumes in naturalistic settings where odour encounters are intermittent, occurring randomly as brief bursts (*Murlis et al., 2000*; *Webster and Weissburg, 2001*). We do note, however, that more stable odour gradients have been mapped to the desert surfaces upon which desert ants forage (*Buehlmann et al., 2015*). Regardless, insect olfactory receptor neurons (ORNs) and projection neurons (PNs) posses adaption (*Kaissling et al., 1987*; *Nagel and Wilson, 2011*) and divisive gain control (*Luo et al., 2010*; *Olsen et al., 2010*; *Gorur-Shandilya et al., 2017*) mechanisms that normalise and smooth noisy olfactory inputs. It is interesting to note that the visual gradients can often present data in a similar noisy fashion (personal observation) and thus raises the question as to whether similar processing steps are applied across modalities. Indeed, this hypothesis is supported by identification of shared early sensory processing principles across sensory modalities (*Wilson, 2013*), especially the vision and olfactory in insects (*Mu et al., 2012*) and mammals (*Cleland, 2010*). Another interesting point is the temporal presentation of information (e.g. continual or discrete) and how this might affect aspects such as optimal integration of cues. We suggest that optimal integration would not be unduly affected as sampling over longer time scales would simply reduce the strength of the more sparse samples cues to the ring attractor. Moreover, there may be benefits in sampling less as it could smooth out local noise in sensory gradients. Investigation of these questions through modelling studies that add more realistic sensory processing in more realistic sensory settings (odour [*Demir et al., 2020*], vision [*Millward et al., 2021*]) is vital to answering these questions.

Despite growing agreement on the functional role of the CX in insect navigation (*Honkanen et al., 2019*; *Hulse et al., 2021*), a number of issues remain. Firstly, as well as innervating the CX, both visual and olfactory cues are also transferred directly to motor centres (*Rayshubskiy, 2020*; *Scaplen et al., 2021*; *Green et al., 2019*) providing redundant information streams. One possibility is that the direct pathways are used for fast reflex-like movements, whereas the CX pathway is responsible for higher-level guidance that requires learning and integration of multiple elemental guidance systems (*Currier et al., 2020*; *Matheson et al., 2021*). This view is consistent with *Steinbeck et al., 2020*, who demonstrate that the LALs, downstream of the CX, possess neural structures well suited to integrating outputs of the fast and the slow pathways. For *Drosophila* larvae, there should be equivalent neural circuitry functioning similarly as the CX involved pathway (probably with the olfactory descending neurons PDM-DN; ; *Gowda et al., 2021*) and direct pathway (probably with odd neurons; *Slater et al., 2015*; *Gowda et al., 2021*). Future work is needed to merge these concepts into a single computational framework. Secondly, there is the question as to whether insects maintain a single or multiple head-direction signals in the PB. In our previous model (*Sun et al., 2020*), we introduced a global celestial compass used by VH and PI behaviours, and a local visual compass for RF. In this study, we relied solely on the global celestial compass, but wind direction sensing from the WPNs is known to feed into the head-direction cells (*Okubo et al., 2020*; *Hulse et al., 2021*), which could facilitate a local compass similar to our previous terrestrial compass. The utility and biological realism of the multi-compass hypothesis deserve further investigation. Thirdly, insects possess an MB in each brain hemisphere, posing the question as to their combined role. *Le Möel and Wystrach, 2020* and *Wystrach et al., 2020* offer the hypothesis that MBs form an opponent memory system that can drive visual RF by balancing the difference in their outputs. This approach can be easily extended to incorporate both attractive and repulsive MB output neurons extending the application space and robustness of navigation. Integration of dual MB inputs represents an obvious next extension of the model presented here. Finally, the model presented here is unique in the format of the sensory data input to the MBs and the behavioural strategies that the MBs generate. Specifically, we propose that the MBs process rotationally invariant but spatially varying cues (e.g. odour and visual familiarity gradients) and are thus responsible for generating gradient ascent/descent behaviours such as VH

and chemotaxis via operant connections to the CX. In contrast, all rotationally varying cues (e.g. wind direction, visual route memories, and celestial compass) innervate the CX directly via alternate pathways (e.g. LAL). This separation of sensory information is fundamental to the flexibility of the model presented to create the array of behaviours presented and offers a testable hypothesis for future work. Such insights will be invaluable for refinement of our understanding of the robust navigation behaviours facilitated by the insect minibrain.

## Materials and methods

All simulations and network models were implemented by Python 3.5 and external libraries *numpy*, *matplotlib*, *scipy*, *opencv*, etc. The source code of the simulation and plotting figures is available via GitHub.

### Odour field

As the basic sensory input, the spatial concentration distribution of the odour field is simulated simply and based on the scaled exponential functions, with required changes according to the wind dynamics.

#### Odour field without wind

For the simulations in the laminar odour environment (i.e. no wind) as that in *Figure 3* (left panel), the landscape of the odour concentration $CON_o$ is modelled for 'volcano' shape:

$$CON_o = \begin{cases} ke^{\tau(r/2-d)} & if \quad d > r/2 \\ ke^{\tau(d-r/2)} & otherwise \end{cases} \tag{1}$$

and for 'linear' shape:

$$CON_o = \begin{cases} ke^{\tau(r/2-d)} & if \quad d > r/2 \\ k - 0.2e^{\tau(d-r/2)} & otherwise \end{cases} \tag{2}$$

where $d$ is the distance from the position $(x, y)$ to the odour source $(x_s, y_s)$. Thus, $d = \sqrt{(x-x_s)^2 + (y-y_s)^2}$ is the scale factor, $r$ is the radius of the odour source, and $\tau$ is the decay factor.

#### Odour field with wind

To simplify the simulation of the odour plume dynamics, all the simulations in this study were conducted under the condition of constant wind speed $u$ and wind direction $\theta_w$, and we assume that the odour plume will ideally flow to the downwind area, that is, the odour concentration in the upwind area will always be zero. The source of the odour constantly emits at the rate $q$. Then the odour concentration at position $(x, y)$ can be calculated by

$$CON_o = \begin{cases} \frac{q}{u\sigma_{xy}\sqrt{2\pi}}e^{-\frac{d^2}{2\pi\sigma_{xy}}} & if \cos\theta > 0 \\ 0 & otherwise \end{cases} \tag{3}$$

where $d = \sqrt{(x-x_s)^2 + (y-y_s)^2}\sin\theta$ is the projected distance from the odour source. And $\sigma_{xy}$ is calculated by $\sigma_{xy} = K_s d$, where $K_s \in [0.5, 0.3, 0.2, 0.15, 0.1]$ is the tuning factor determined by the stability of the odour. And $\theta$ is the angle between the vector pointing from the position to the source and the wind direction, so can be computed by

$$\theta = \arccos \frac{(x-x_s)(u\cos\theta_w)+(y-y_s)(u\sin\theta_w)}{\sqrt{(x-x_s)^2+(y-y_s)^2}u} \tag{4}$$

### Neural model

We use the simple firing rate to model the neurons in the proposed networks, where the output firing rate $C$ is a sigmoid function of the input $I$ if there is no special note. In the following descriptions

and formulas, a subscript is used to represent the layers or name of the neuron while the superscript is used to represent the value at a specific time or with a specific index.

## Current heading

In our previous model, there are two compass references derived from different sensory information (**Sun et al., 2020**), but in this paper, only the global compass (i.e. the activation of I-TB1/Δ7 neuron) is used here because navigation behaviours reproduced in this study are all assumed using the global compass as the external direction reference. For the details of the modelling of global current heading ($I^{t,j}_{I-TB1}$), see our previous paper (**Sun et al., 2020**).

## Steering circuit

The steering neurons (the same as previous paper [**Sun et al., 2020**] but presented here for convenience), that is, CPU1 neurons ($C^i_{CPU1}, i = 0, 1, 2...15$) receive excitatory inputs from the desired heading ($C^i_{DH}, i = 0, 1, 2...15$) and inhibitory inputs from the current heading ($C_{CH}, i = 0, 1, 2...15$) to generate the turning signal:

$$C^i_{ST} = C^i_{DH} - C^i_{CH} \qquad i = 0, 1, ...15 \tag{5}$$

The turning angle is determined by the difference of the activation summations between left ($i = 0, 1, 2...7$) and right ($i = 8, 9, 10...15$) set of CPU1 neurons:

$$\theta_M = k_{motor}(\textstyle\sum_{i=0}^{7} C_{CPU1} - \sum_{i=8}^{15} C_{CPU1}) \tag{6}$$

## Upwind direction encoding

The upwind direction is decoded as the activation of UW neurons copied and shifted from heading neurons (I-TB1), and the value of this shifting is determined by the angular difference between the current heading ($\theta_h$) and wind direction ($\theta_w$) encoded by the firing rate of WPN. And the value of WPN is defined as the difference of the antennal deflection encoded by B1 and APNs as

$$C_{WPN} = C_{APN} - C_{B1} = \sin(\theta_w - \theta_h + \pi) - \sin(-(\theta_w - \theta_h + \pi)) \tag{7}$$

Then population activation of upwind direction neurons (UW) can be calculated by

$$C_{UW} == C^j_{I-TB1}, j = \begin{cases} i + offset & if \quad i + offset \leq 7 \\ i + offset - 7 & otherwise \end{cases} \tag{8}$$

## Fly: ON and OFF responses-based switching circuit

Different navigation strategies will dominate the motor system according to the sensory inputs, that is, in this study, the change of perceived odour concentration. This coordination is modelled as a contextual switching that is very similar to the mechanism with SN1 and SN2 neuron involved in our previous model (**Sun et al., 2020**) to define the final output of odour navigation ($C_{ON}$):

$$C^i_{ON} = \begin{cases} C^i_{chemo} & if \quad OFF\ response \\ C^i_{anemo} & if \quad ON\ response \end{cases} \tag{9}$$

And how the sensory information determines the response is shown in **Table 1**, where random means no reliable sensory input is available, and the agent will move forward to a random direction.

**Table 1.** 'Truth table' of the ON and OFF responses of the modelled fly odour navigation.

Columns list the state of sensed odour concentration while rows indicate the state of the changing of odour concentration.

|  | $< Thr_{off}$ | $> Thr_{off}$ $< Thr_{on}$ | $> Thr_{on}$ |
|---|---|---|---|
| $< Thr_o$ | Random | Random | ON |
| $> Thr_o$ | OFF | ON | ON |

## OFF response: Chemotaxis

The chemotaxis model is adapted from the previous VH model (*Sun et al., 2020*) by changing the change of visual familiarity signal from the MBON ($\Delta C_{MBON}$) to the change of the odour concentration to determine the shifting value, thus the desired heading of chemotaxis is

$$C^i_{chemo} = C^j_{I-TB1}, j = \begin{cases} i + \textit{offset} & if \quad i + \textit{offset} \leq 7 \\ i + \textit{offset} - 7 & otherwise \end{cases} \qquad i = 0, 1, ...7 \qquad (10)$$

Note that in our previous visual navigation model (*Sun et al., 2020*) $i, j$ both were integers for the ease of computing, thus, the shifting resolution is 45°, but here to more accurately model the desired heading and to achieve better performance, the shifting resolution was set to be 4.5° by interpolating neuron activation of I-TB1 from 8 to 80, then down-sampling to 8 to generate the shifted desired heading.

The relationship between $\Delta C_o$ and *offset* is shown as follows:

$$\textit{offset} = \begin{cases} 0 & if \quad \Delta C_o < 0 \\ \min(\lfloor k_{chemo} \Delta C_o \rfloor, 3) & otherwise \end{cases} \qquad (11)$$

Then the desired heading of OH will be fed into the steering circuit to compare with the current heading to generate the motor command.

## ON response: Odour-gated anemotaxis

As shown in *Table 1*, when the ON response is determined, the agent will follow the upwind direction, thus the desired heading input to steering circuit should be the upwind direction encoded by UM neuron (*Equation 8*):

$$C^i_{anemo} = C^i_{UW} \qquad (12)$$

### Ants: Integration with PI

The modelling of ants' odour navigation integrated with PI can be regarded as the extension of the fly's odour navigation and an application of the unified model. Specifically, the final output of olfactory navigation is determined by the ON and OFF responses (see *Table 1*), and then is integrated with PI via RA like that in the optimal integration of PI and VH:

$$\tau \frac{dC_{IN}}{dt} = -C_{IN} + g\left(\sum_{j=1}^n W^{ji}_{E2E} C^j_{IN} + X^i_1 + X^i_2 + W_{I2E} C_{UI}\right) \qquad i = 0, 1, ...7. \qquad (13)$$

where $W^{ji}_{E2E}$ is the recurrent connections from $j^{th}$ neuron to $i^{th}$ neuron, $g(x)$ is the activation function that provides the non-linear property of the neuron:

$$g(c) = max(0, \rho + c) \qquad (14)$$

where $\rho$ denotes the offset of the function. Thus $X1$ should be

$$X^i_1 = C^i_{PI} \qquad i = 0, 1, ...7 \qquad (15)$$

and $X2$ in *Equation 12* should be

$$X^i_2 = \begin{cases} k_o CON_o C^i_{OH} & if \quad OFF\ response \\ k_o CON_o C^i_{anemo} & if \quad ON\ response \end{cases} \qquad (16)$$

Then the output of optimal integration (OI) of the RA acts as the only desired heading input to the steering circuit:

$$\begin{cases} C^{0-7}_{DH} = C_{OI} W_{DH2CPU1L} \\ C^{8-15}_{DH} = C_{OI} W_{DH2CPU1R} \end{cases} \qquad (17)$$

**Table 2.** The detailed parameter settings for the simulations in this study.

| | | Fly – chemotaxis | | Fly – anemotaxis | Fly – integrated | Ant – integrated |
|---|---|---|---|---|---|---|
| | | Volcano | Linear | | | |
| | $k$ | 10 | 10 | | | |
| | $\tau$ | 0.1 | 0.1 | | | |
| | $r$ | 6 | 6 | / | | |
| Odour | $q$ | / | / | 10.0 | 10.0 | 20 |
| | $u$ | | | 10.0 | 10.0 | 10.0 |
| Wind | $w_\theta$ | No wind | | $-\pi/2$ | $-\pi/2$ | $\pi$ |
| | $Thr_o$ | | | | 0.001 | 1.2 |
| | $Thr_{on}$ | | | | 0.02 | 0.5 |
| | $Thr_{off}$ | | | | -0.0002 | -0.0002 |
| | $k_\circ$ | / | | / | / | 0.5 |
| | $k_{chemo}$ | 100.0 | 100.0 | / | 100.0 | 100.0 |
| | $k_{motor}$ | 1.0 | 1.0 | 1.5 | 1.5 | 1.0 |
| | $S_L$ | 0.02 | 0.02 | 0.4 | 0.4 | 0.05 |
| Model and simulation | Heading | Random | Random | Random | Random | $0 - 2\pi$ |

Only the global compass is needed in this study's modelling. Thus the input of current heading will always be the excitation of the I-TB1 neuron:

$$\begin{cases} C_{CH}^{0-7} = C_{I-TB1} \\ C_{CH}^{8-15} = C_{I-TB1} \end{cases} \tag{18}$$

The output of the steering circuit (i.e. the summed activation of the left and right CPU1 neurons) is used to generate the turning command in the way that is same as *Equation 6*.

## Simulations
In all simulations, at each time step, the simulated agent (walking fly or ant) will sense the odour sensory based on its current location and then update neural activation to generate the desired moving direction and finally move one step to that direction. *Equation 6* gives the turning angle of the agent, thus the instantaneous 'velocity' (*v*) at every step can be computed by

$$\boldsymbol{v}^t = S_L[\cos\theta_M^t, \sin\theta_M^t] \tag{19}$$

where $S_L$ is the step length in centimetres. Note that we have not defined the time accuracy for every step of the simulations, thus the unit of the velocity in this implementation is *cm/step* rather than *cm/s*. Then the position of agent $\boldsymbol{P}^{t+1}$ in the Cartesian coordinates for the next time step is updated by

$$\boldsymbol{P}^{t+1} = \boldsymbol{P}^t + \boldsymbol{v}^t \tag{20}$$

The positions of odour sources in all simulations are all set to $(0,0)$, that is, $x_s = 0, y_s = 0$. Other main parameters are listed in *Table 2*. Note that in each simulation the speed of agent is set constant.

### Fly: Chemotaxis
To test the performances of the chemotaxis behaviour, five simulated agents with randomly generated heading direction start from five randomly generated locations in the zone of $(-12 < x < 12, -12 < y < 12)$, and are then driven by the model for 1500 steps. Then we ran this

simulation for four times in two different odour landscapes ('volcano' and 'linear') to get the results shown in *Figure 3* (right panel) and *Figure 4*.

### Fly: Anemotaxis

To reproduce the behavioural data in *Álvarez-Salvado et al., 2018*, the odour was only set ON during the second a quarter of total time (e.g. if the agent is set to run 200 steps, then the odour ON time will be in 50–100 steps). Four agents with randomly generated heading start from randomly generated locations in the zone of $(-1.5 < x < 1.5, -13 < y < -5)$, and are then guided by the model to run 200 steps. The simulation was conducted for five times.

### Fly: Integrated ON and OFF response

The whole simulation settings are the same as that in the last section except for some model parameters listed in *Table 2* as this simulation is conducted to verify the integrated model.

### Ants: Odour navigation integrated with PI

To reproduce the behavioural data in *Buehlmann et al., 2012*, we first generated PI memory encoding the home vector with 10 m length and $\pi/2$ direction. Then at each release point $((-1.5, -10)$ and $(1.5, -10))$, we released 10 simulated full-vector (10-m-long and pointing to $\pi/2$) ants with different initial headings sampled uniformly from $0 - 2\pi$ (see also *Table 2*). Note that the simulation settings with/without additional odour plume diffused by conspecific nest are identical so the list as one column in *Table 2*.

### Ants: Wind compensation and backtracking

The quick implementations of using 'copy-and-shift' mechanism to model the wind compensation and backtracking behaviour follow the same step: first, generate the desired headings by shifting the current heading by the WPN activation for the wind compensation and by 180° for backtracking, respectively; second, release the simulated ant at the same releasing point but with random headings (uniform distribution in $0 - 2\pi$). Motion-related parameters are set identically as that of those mentioned in the section 'Ants: odour navigation integrated with PI'.

## Acknowledgements

This research has received funding from the European Union's Horizon 2020 research and innovation programme under the Marie Sklodowska-Curie grant agreement no. 778062, ULTRACEPT, and no. 691154, STEP2DYNA.

## Additional information

### Funding

| Funder | Grant reference number | Author |
|---|---|---|
| EU Horizon 2020 Framework Program | ULTRACEPT 778062 | Xuelong Sun Shigang Yue |

The funders had no role in study design, data collection and interpretation, or the decision to submit the work for publication.

### Author contributions

Xuelong Sun, Conceptualization, Data curation, Formal analysis, Investigation, Methodology, Resources, Software, Validation, Visualization, Writing – original draft, Writing – review and editing; Shigang Yue, Conceptualization, Funding acquisition, Investigation, Project administration, Supervision, Writing – review and editing; Michael Mangan, Conceptualization, Formal analysis, Investigation, Methodology, Supervision, Validation, Visualization, Writing – original draft, Writing – review and editing

### Author ORCIDs
Xuelong Sun  http://orcid.org/0000-0001-9035-5523

### Decision letter and Author response
Decision letter https://doi.org/10.7554/eLife.73077.sa1
Author response https://doi.org/10.7554/eLife.73077.sa2

## Additional files

### Supplementary files
• Transparent reporting form

### Data availability
The current manuscript is a computational study, so no data have been generated for this manuscript. Modelling code is uploaded as Source Code File and is also available via Github (https://github.com/XuelongSun/insectNavigationCX, copy archived at https://archive.softwareheritage.org/swh:1:rev:ec0d6943e09df2a685f8b1382475c523375352c3).

The following dataset was generated:

| Author(s) | Year | Dataset title | Dataset URL | Database and Identifier |
| --- | --- | --- | --- | --- |
| Sun X | 2021 | How the insect central complex could coordinate multimodal navigation | https://github.com/XuelongSun/insectNavigationCX | Github, insectNavigationCX |

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
