## [Editor Report]

This *eLife* Advance by Sun et al. expands on a previous publication in 2020 which developed and outlined a model for biologically inspired visual navigation circuit. Here they include odour and wind input and successfully recreate complex multimodal behavioural observations in ants, illustrating that the model is suited to generate viable hypotheses about circuit-level implementation of navigational control networks in insects in response to varied sensory inputs.

---

## [Decision Letter]

**Decision letter after peer review:**

Thank you for submitting your article "How the insect central complex could coordinate multimodal navigation" for consideration by *eLife*. Your article has been reviewed by 3 peer reviewers, and the evaluation has been overseen by Mani Ramaswami as Reviewing Editor and Ronald Calabrese as the Senior Editor. The following individual involved in review of your submission has agreed to reveal their identity: Stanley Heinze (Reviewer #1).

Essential revisions:

While the work is significant, valuable and of general interest, several key issues that need to be addressed prior to publication are enumerated below.

1) The authors integrate odor based and visual navigation behavior in the same model by simply swapping out the sensory apparatus and early stage processing pathways. While the principal approach is not unreasonable, it is possible that this is an over-simplification. Odor information reaches the antenna in discrete packages (pulses resulting from pockets/filaments of odor molecules suspended in an air stream), while visual information is sampled continuously. Yet the model appears to treat both equivalently and updates each sensory input with every step of the model? Would the model still perform well, if the olfactory input were present intermittently? Could odor concentrations be varied such that odor packages are received only occasionally? How would that affect optimal cue integration? Note that the these are is the likely precise nature of odor stimuli in air rather the "pure" odor gradients used in Matthieu Louis's fly larval experiments. If the authors agree that this is a relevant point, potential consequences in the paper should be discussed, ideally, by includung modeling data showing the impact of different temporal structure of stimuli.

2) The authors postulate a ring attractor circuit in the FB of the CX and refer to the fly connectome (Hulse et al.,) for support. However, that paper explicitly states (while referring to path integration memory): "… Solutions like these seem unlikely to be implemented by the FB, since they require FB-centered attractors with circuit motifs for shifting the bump, similar to those found in the EB-PB attractor, which we see no evidence for in the FB network. (Hulse et al., 2021)" There is clearly the possibility that the fly connectome has missed footprints of such a circuit, or that ants and bees contain circuits that flies are lacking. This should be made explicit in the paper, to not give the impression that FB ring attractors have been found.

3) In line with comment 2, the model might be better called biologically inspired or plausible rather than realistic, simply to avoid that readers take the model as biological reality.

4) The authors introduce the odor input to the model by stating that odors reach the CX via known pathways, which can be swapped out for the visual inputs. This implies that processing is similar in these pathways and that it is indeed sensory information that reaches the CX. However, for olfactory information, the downstream targets of the lateral horn are not well described and, to my knowledge, do not include any CX neurons. Secondly, the output neurons of the MB are no longer sensory, but encode stimulus valence. The equivalence suggested by the text and implied in the model, is thus not biologically realistic. These simplifications should be more explicit in the text.

5) The pathway described for wind processing seems not fully correct (line 112). The AMMC is the target region of the antennal sensory neurons. Their targets then project to the wedge, from where neurons connect to the LAL and CX. In the text, the order is wedge first, then AMMC, LAL and CX. Please re-check and rephrase.

6) As the authors mention in Discussion, there are known to be direct connections from the MB to the LAL that likely mediate learned odor attraction/avoidance. This is in addition to connections to motor centers from pathways originating in the lateral horn that likely help with navigation behaviors triggered by innate odor preference. Is it the authors belief that all chemotaxis would go through the CX, as they appear to be suggesting in this integrated model (e.g., in lines 101-102)? Why? This needs to be much better motivated and discussed, particularly because *Drosophila larvae*, which are mentioned frequently in this study, do not even have what we would think of as a CX, and yet seem to perform pure chemotaxis much like the model agent does.

7) It's true that the WPN neurons do feed into the NO and then the FB, as the authors assume. However, the WPN neuron is also an input to the ER (ring) neurons that innervate the ellipsoid body and connect to the compass/HD system (Hulse et al., 2020). That is, wind direction is not just a self-motion cue, but is something that the HD system can directly tether to and that can help the compass integrate multiple sensory cues through plasticity in the EB. This merits some discussion, if not being added to the model, because it may affect how exactly the copy-and-shift mechanism should work with and without wind. This is also related to the switch between the first and current paper when it comes to model implementations: assuming a single "global" compass here versus assuming two different compass references in the previous model.

8) It would help if the figures, figure legends and Results are made to be more self-explanatory rather than requiring the reader to repeatedly refer to the earlier paper.

---

## [Author Response]

Essential revisions:While the work is significant, valuable and of general interest, several key issues that need to be addressed prior to publication are enumerated below.1) The authors integrate odor based and visual navigation behavior in the same model by simply swapping out the sensory apparatus and early stage processing pathways. While the principal approach is not unreasonable, it is possible that this is an over-simplification. Odor information reaches the antenna in discrete packages (pulses resulting from pockets/filaments of odor molecules suspended in an air stream), while visual information is sampled continuously. Yet the model appears to treat both equivalently and updates each sensory input with every step of the model? Would the model still perform well, if the olfactory input were present intermittently? Could odor concentrations be varied such that odor packages are received only occasionally? How would that affect optimal cue integration? Note that the these are is the likely precise nature of odor stimuli in air rather the "pure" odor gradients used in Matthieu Louis's fly larval experiments. If the authors agree that this is a relevant point, potential consequences in the paper should be discussed, ideally, by includung modeling data showing the impact of different temporal structure of stimuli.

These points are highly relevant and very interesting. Thank you. Firstly, we agree that the model is an abstraction of the olfactory processing mechanism. However, we believe that this is warranted as our primary aim was to show (a) that problems in different sensory domains share features (e.g., odour and visual gradients), and thus (b) can be solved by the same central brain structures without significant changes. That is, to focus on the inner-brain processing, we omitted the modelling of early-stage olfactory processing happens in the olfactory receptor neurons (ORNs) and projection neurons (PNs). Future models could include information processing features such as adaption (Kaissling et al., 1987; Nagel and Wilson, 2011), divisive gain control (Luo et al., 2010; Olsen et al., 2010; Gorur-Shandilya et al., 2017), etc., which normalises and smooth complex data in natural odour plumes. It is interesting to consider whether similar mechanisms could address similar problems observed in noisy visual gradients.

Indeed, studies have shown that early sensory processing shares some fundamental principles across sensory modalities (Wilson, 2013), especially the vision and olfactory in insects (Mu et al., 2012) and mammals (Cleland, 2010), which provides supports for our assumption that visual and olfactory sensory could be processed in a similarly way serving as the sensory basis of navigation behaviours.

Modelling the sensory processing is beyond the scope of this work however we have added significant amount of text to the Discussion section (revised version Lines 264-286) outlining hypothesis and future works.

2) The authors postulate a ring attractor circuit in the FB of the CX and refer to the fly connectome (Hulse et al.,) for support. However, that paper explicitly states (while referring to path integration memory): "… Solutions like these seem unlikely to be implemented by the FB, since they require FB-centered attractors with circuit motifs for shifting the bump, similar to those found in the EB-PB attractor, which we see no evidence for in the FB network. (Hulse et al., 2021)" There is clearly the possibility that the fly connectome has missed footprints of such a circuit, or that ants and bees contain circuits that flies are lacking. This should be made explicit in the paper, to not give the impression that FB ring attractors have been found.

Thanks for this useful feedback. As suggested, we have updated the text in the Discussion section (revised version Line 256-263) to address this issue. Additionally, based on the new data published very recently in Sayre et al., (2021), we added new hypotheses regarding the mapping from our model to the real neurons.

3) In line with comment 2, the model might be better called biologically inspired or plausible rather than realistic, simply to avoid that readers take the model as biological reality.

We have changed the phrase 'biologically realistic' to 'biologically plausible' or "biologically constrained" throughout the text where there is lack of strong biological evidence.

4) The authors introduce the odor input to the model by stating that odors reach the CX via known pathways, which can be swapped out for the visual inputs. This implies that processing is similar in these pathways and that it is indeed sensory information that reaches the CX. However, for olfactory information, the downstream targets of the lateral horn are not well described and, to my knowledge, do not include any CX neurons. Secondly, the output neurons of the MB are no longer sensory, but encode stimulus valence. The equivalence suggested by the text and implied in the model, is thus not biologically realistic. These simplifications should be more explicit in the text.

Regarding the first point, there is some evidence to suggest that the lateral horn transmits to the CX through indirect pathways (Dolan et al., 2019; Matheson et al., 2021). To make this clearer we now show indirect connections in figures with dashed lines, and direct pathways in solid lines.

Regarding the second point, we appreciate the clarification in terminology and have updated throughout. Besides, we also added a sentence in the Result section (revised version Line 100-102) to further clarify this.

5) The pathway described for wind processing seems not fully correct (line 112). The AMMC is the target region of the antennal sensory neurons. Their targets then project to the wedge, from where neurons connect to the LAL and CX. In the text, the order is wedge first, then AMMC, LAL and CX. Please re-check and rephrase.

This is indeed an error, and we have amended in the Results section (revised version Line 118-120). Thanks for spotting this.

6) As the authors mention in Discussion, there are known to be direct connections from the MB to the LAL that likely mediate learned odor attraction/avoidance. This is in addition to connections to motor centers from pathways originating in the lateral horn that likely help with navigation behaviors triggered by innate odor preference. Is it the authors belief that all chemotaxis would go through the CX, as they appear to be suggesting in this integrated model (e.g., in lines 101-102)? Why? This needs to be much better motivated and discussed, particularly because *Drosophila larvae*, which are mentioned frequently in this study, do not even have what we would think of as a CX, and yet seem to perform pure chemotaxis much like the model agent does.

Thanks for this useful comment. We don't believe that all the chemotaxis should go through the CX. Instead, what we aim to demonstrate is that our model of the CX could facilitate chemotaxis-like behaviours as these behaviours share the identical sensory-motor mechanism with other navigation behaviours (e.g., visual homing) that can be uniformly explained by our model. Moreover, the CX can coordinate disparate navigation behaviours in a principled way that is likely important for solving complex tasks in complex real world scenarios.

Our Discussion section already highlighted pathways bypassing the CX to the motor centre responsible for the fast reacting behaviours (e.g. reflex). Indeed, *Drosophila* larvae do not have the CX, but if they are capable of doing memory-based odour navigation, then we argue that there must be equivalent neural circuity functioning similarly as the CX (probably the olfactory descending neurons PDM-DN (Ibrahim et al., 2018; Gowda et al., 2021)), otherwise they apply innate-driven chemotaxis without the involvement of higher brain regions (probably purely through Odd neurons (Slater et al., 2015; Gowda et al., 2021)). To address the reviewer’s concerns, we have: (1) added several sentences to discuss this in Results section (revised version Line 105-111); (2) expanded the Discussion section related to this point (revised version Line 295-299); and (3) inserted a new panel to Figure 2 (revised version) depicting the possible map of the functional roles of the brain region in the larva's central nervous systems (CNS).

7) It's true that the WPN neurons do feed into the NO and then the FB, as the authors assume. However, the WPN neuron is also an input to the ER (ring) neurons that innervate the ellipsoid body and connect to the compass/HD system (Hulse et al., 2020). That is, wind direction is not just a self-motion cue, but is something that the HD system can directly tether to and that can help the compass integrate multiple sensory cues through plasticity in the EB. This merits some discussion, if not being added to the model, because it may affect how exactly the copy-and-shift mechanism should work with and without wind. This is also related to the switch between the first and current paper when it comes to model implementations: assuming a single "global" compass here versus assuming two different compass references in the previous model.

This feedback is great as it raises the key question as to whether insects possess a single 'sense of direction' or whether they possess multiple compass systems. In our previous work we had two compass systems: a local compass used for route following, and a global (celestial compass) used for visual homing and path integration. In this work, we use only a global compass for reference. We agree that this is a very interesting topic, which is novel and must be tackled delicately. Indeed, it is our intention to follow this work with a full paper dedicated to this issue. Thus, for this paper we have instead added text to the Discussion section (revised version Line 300-306) that clarifies the idea of multiple compass systems and outlines potential avenues for future research.

8) It would help if the figures, figure legends and Results are made to be more self-explanatory rather than requiring the reader to repeatedly refer to the earlier paper.

Thanks for this feedback, to make this paper more self-explanatory, we have: (1) Carefully re-checked and updated the captions and legends in all figures; (2) Added an overview figure at the very beginning of this paper to show the main components published in our previous papers that we would apply in this study, e.g., steering circuit, ring attractor for optimal integration, switching circuit for contextual switching, etc. (see the new Figure 1).